# FineMoGen: Fine-Grained Spatio-Temporal Motion Generation and Editing

**Mingyuan Zhang**[1]  **Huirong Li**[1]  **Zhongang Cai**[1,2]  **Jiawei Ren**[1]  **Lei Yang**[2]  **Ziwei Liu**[1]

[1] S-Lab, Nanyang Technological University, [2] SenseTime Research

## Abstract

Text-driven motion generation has achieved substantial progress with the emergence of diffusion models. However, existing methods still struggle to generate complex motion sequences that correspond to fine-grained descriptions, depicting detailed and accurate spatio-temporal actions. This lack of fine controllability limits the usage of motion generation to a larger audience. To tackle these challenges, we present **FineMoGen**, a diffusion-based motion generation and editing framework that can synthesize fine-grained motions, with spatial-temporal composition to the user instructions. Specifically, FineMoGen builds upon diffusion model with a novel transformer architecture dubbed Spatio-Temporal Mixture Attention (**SAMI**). SAMI optimizes the generation of the global attention template from two perspectives: **1)** explicitly modeling the constraints of spatio-temporal composition; and **2)** utilizing sparsely-activated mixture-of-experts to adaptively extract fine-grained features. To facilitate a large-scale study on this new fine-grained motion generation task, we contribute the **HuMMan-MoGen** dataset, which consists of 2,968 videos and 102,336 fine-grained spatio-temporal descriptions. Extensive experiments validate that FineMoGen exhibits superior motion generation quality over state-of-the-art methods. Notably, FineMoGen further enables zero-shot motion editing capabilities with the aid of modern large language models (LLM), which faithfully manipulates motion sequences with fine-grained instructions. Project Page: https://mingyuan-zhang.github.io/projects/FineMoGen.html.

## 1 Introduction

While traditional text-driven motion generation tasks have empowered models to generate motion sequences based on text, two issues still persist: 1) Users can hardly control the generated actions with fine-grained spatio-temporal details; 2) Users struggle to further edit the already generated animations. These two drawbacks significantly limit the application scenarios of these algorithms. Therefore, in this paper, we propose a new task, Fine-grained Spatio-temporal Motion Generation and Editing. We anticipate that the new algorithm framework derived from this task will better accept detailed commands from users and interactively optimize the generated action sequences according to users' further needs. This would thus allow text-driven motion generation technology to be utilized more widely by the general public.

Previous work has made some initial strides in fine-grained motion generation, but often ends up in one of two extremes: 1) it heavily relies on full supervision, producing good generation results but requiring extensive and detailed annotation [11]; 2) it fully depends on zero-shot inference, allowing use in various scenarios, but the generated results often contain many artifacts [20, 18]. These work have not deeply explored the fine-grained correspondence between text and motion. This fine-grained relationship is represented spatially as the coordination of body parts and temporally as semantic consistency. With the help of such granular relationships, a sequence of motions that is spatially reasonable and temporally semantically consistent can be composed from the bottom up. To better

37th Conference on Neural Information Processing Systems (NeurIPS 2023).

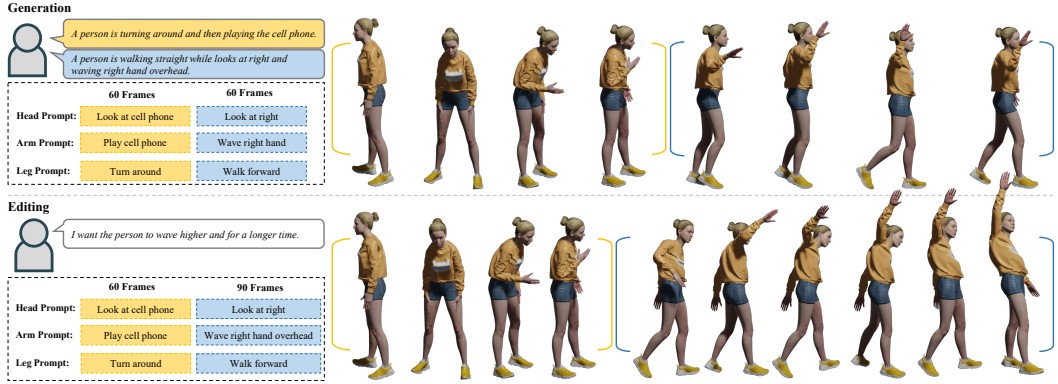

Figure 1: FineMoGen is a motion diffusion model that can accept fine-grained spatio-temporal descriptions. The synthesized motion sequences are natural and consistent with the given conditions. With the assistance of Large Language Model (LLM), users can interactively edit the generated sequence.

capture this correlation, we propose a new framework, FineMoGen. We explicitly model the influence of space and time in the attention mechanism we use, which not only improves the performance in full-supervise scenarios, but also mines spatio-temporal information from standard text-driven motion generation datasets, achieving better results in zero-shot scenarios. Specifically, we have made two modifications based on the Efficient Attention [16] used in previous work [20, 21]: 1) We delve deeper into the properties of Efficient Attention used therein, proposing a spatio-temporal decoupled method for explicit modelling. This not only supports both single text condition and fine-grained text condition formats during training, but also allows for the introduction of more conditions during testing; 2) We employ Sparsely-activated Mixture-of-Experts to adaptively provide the most suitable information, thereby optimizing feature quality.

To thoroughly evaluate our proposed algorithm, we have conducted extensive experimental analysis on existing standard datasets such as HumanML3D [6], KIT-ML [12], and BABEL [13]. However, the existing datasets only provide rough description of human motion as a whole, without fine-grained descriptions of body parts and temporal decomposition. To promote fine-grained motion synthesis, we selected the comprehensive HuMMan [3] dataset and provided it with fine-grained text annotations. Specifically, we divided a long motion sequence into several different motion stages. For each stage we provided an overall text description and detailed descriptions for seven different body parts. This dataset contains a total of 2,968 videos and 102,336 text annotations, which can effectively support future research on fine-grained spatio-temporal motion generation algorithms. Experimental results show that our method has achieved state-of-the-art levels on these benchmarks. Visualization results further demonstrate that our method can generate more natural motion sequences that are more consistent with the given commands, whether in full-supervise scenarios or zero-shot scenarios.

Building on the foundation of fine-grained spatio-temporal motion generation, we further explored its integration with large language models (LLMs). Our framework allows users to interact with the LLM using natural language instructions, which in turn automatically modify the fine-grained descriptions. This enables our algorithm to adjust the content and style of the generated motion sequences according to user commands, enhancing interaction efficiency and extending application possibilities.

Our work has made the following three major contributions:

1. We have proposed the first framework capable of fine-grained motion generation and editing that applies to both zero-shot and fully-supervised scenarios.

2. We have delved deeply into the attention technology used in the motion diffusion model, proposing a spatio-temporal decoupled method for generating global templates, enhanced by the use of Mixture of Experts.

3. We have established a large-scale dataset with fine-grained spatio-temporal text annotations, HuMMan-MoGen, to facilitate future research in this direction.

## 2 Related Works

### 2.1 Text-driven Motion Generation

Recent years have seen significant progress in text-driven motion generation. Traditional research in this field often requires the model to learn a multimodal joint space comprising both text descriptions and motion sequences. For instance, JL2P [1] integrates language and pose into a joint embedding space. Ghosh *et al.* [5] strive to learn a joint-level mapping between language and pose, breaking down the pose embedding into two separate body part embeddings. MotionCLIP [17] also aligns text and motion in a shared embedding space. By leveraging the shared text-image latent space learned by CLIP, MotionCLIP is capable of generating out-of-distribution and stylized motions. In the pursuit of generating motion sequences with high diversity, several works have introduced variational mechanisms. For instance, TEMOS [11] employs transformer-based VAEs to produce motion sequences conditioned on text descriptions. Guo *et al.* [6] propose an auto-regressive conditional VAE that generates a short clip in a recursive manner. TM2T[8] leverages a neural model for machine translation to facilitate the mapping between text and motion. T2M-GPT[19] reframes the problem of text-driven motion generation as a next-index prediction task by mapping a motion sequence to a sequence of indices. Recently, diffusion-based generative models have shown impressive performance on leading benchmarks for text-to-motion task. MotionDiffuse [20], MDM [18], FLAME [10] are the first attempts to apply diffusion model into text-driven motion generation field. MLD [4] further exploit the usage of latent diffusion model and achieve a better performance. ReMoDiffuse [21] integrates the retrieval technique to motion generation pipeline. The retrieved samples provide informative knowledge for the generation process and thus can generate more natural motion sequences.

### 2.2 Spatio-Temporal Motion Composition

TEACH [2], based on the TEMOS [11] algorithm, extracts features from previously generated motion sequences and combines them with text features to obtain a latent code for motion reconstruction. This method is simple and effective, but it must be established under a full-supervised scenario to fit the adjacency relationship between motions in adjacent labels, limiting its usage in broader scenarios. Furthermore, all conditions are compressed into a single latent code, which can easily lose low-level kinematic characteristics. On the other hand, diffusion model-based motion generative models can leverage the advantages of diffusion models for zero-shot spatio-temporal composition [20, 18, 15]. Going further, MotionDiffuse [20] proposes a smoothing term to make the generated motions appear more natural. PriorMDM [15] present a two-stage generation scheme as an inference trick to generate smoother motion sequence. However, due to the lack of effective supervised learning, these types of generated motions often contain many artifacts, such as abrupt changes in speed. These two predicaments mean that there is currently no appropriate method that can truly meet user needs. In this paper, we introduce a novel attention mechanism that is capable of not only uncovering spatio-temporal correlations in standard text-to-motion benchmarks but also more effectively learning transition schemes in a fully-supervised context.

## 3 Methodology

In this section, we will thoroughly introduce our proposed method, FineMoGen. We first present our overall framework in Section 3.1. Then we define the problem of Fine-grained Spatio-Temporal Generation and list the preliminaries used in our method, including the formulas for the diffusion model and the network architecture in Section 3.2. We then detail our proposed **Sp**A**tio-Temporal **MI**xture Attention (**SAMI**), in Section 3.3. In Section 3.4, we explain how we have implemented zero-shot motion editing by leveraging the definition of fine-grained description. Finally, in Section 3.5, we discuss the methods used in the training and testing phases.

### 3.1 Framework Overview

Figure 2 illustrates the entire process of our proposed FineMoGen, a transformer-based motion diffusion model. As for fine-grained spatio-temporal motion generation, the fine-grained descriptions are first passed through a frozen-CLIP model to generate a text feature matrix. This feature matrix is

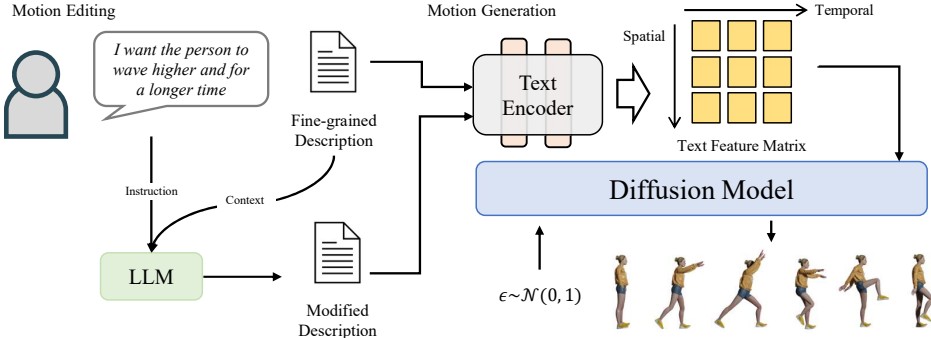

Figure 2: **An overview of FineMoGen**. As for the motion generation task, the fine-grained descriptions are first processed by a text encoder. A text feature matrix can be acquired, and then sent to a diffusion model-based motion generative network to generate corresponding motion sequence. For the editing purpose, LLM is used to interact with users and modify the fine-grained description accordingly.

then fed into the transformer, where it influences the generation of various global templates within the SAMI mechanism. During the editing phase, users interact with a large language model (LLM) to efficiently modify the existing fine-grained descriptions. After updating the conditions, FineMoGen generates new motion sequences that better align with the users' expectations.

## 3.2 Preliminaries

**Problem Definition.** Similar to the standard text-driven motion generation task, the Fine-grained Spatio-temporal motion generation also demands the model to generate realistic action sequences consistent with the given text description. The difference lies in that the latter provides multiple fine-grained descriptions with their respective time ranges and body part scopes. Formally, in Fine-grained Spatio-temporal motion generation task, we define the annotation format as a description matrix $\text{Text}_{i,j}, i \in [1, N_T], j \in [1, N_S]$, where $N_T$ and $N_S$ indicate the number of stages and number of body parts we divide. Accordingly, the motion sequences we generate need to satisfy all constraints, i.e., the entire action sequence $\Theta$ can be divided into $N_T$ stages, with each stage containing $N_S$ body parts. We expect that the motion sequence $\Theta_{i,j}$, corresponding to the $j$th body part in the $i$th stage, aligns with the given text description $\text{Text}_{i,j}$. Additionally, we aim for the generated motions to appear natural as a whole. This means that we need to ensure smooth coordination among different body parts' movements and achieve seamless transitions between different time segments.

**Diffusion Model.** In line with the approach of ReMoDiffuse [21], we have developed a transformer-based diffusion model specifically to tackle the fine-grained spatio-temporal motion generation task. The diffusion model integrates two key components for optimal generation quality: a diffusion process and a corresponding reverse process.

*Diffusion Process* is modeled as a $T$-step Markov chain, where the noised sequences $\mathbf{x}_1, \ldots, \mathbf{x}_T$ are distortions of the real data $\mathbf{x}_0 \sim q(\mathbf{x}_0)$. The intermediate sequence $\mathbf{x}_t$ can be represented by a series of poses $\theta_i \in \mathbb{R}^D, i = 1, 2, \ldots, N_m$, where $D$ is the dimensionality of the pose representation and $N_m$ is the number of frames. Each diffusion step inject Gaussian noises to the data following the formulation: $q(\mathbf{x}_t|\mathbf{x}_{t-1}) := \mathcal{N}(\mathbf{x}_t; \sqrt{1 - \beta_t}\mathbf{x}_{t-1}, \beta_t\mathbf{I})$. An efficient approximation from Ho *et al.* [9] simplify the multi-step diffusion process by $\mathbf{x}_t := \sqrt{\bar{\alpha}_t}\mathbf{x}_0 + \sqrt{1 - \bar{\alpha}_t}\epsilon$, where $\alpha_t := 1 - \beta_t$ and $\bar{\alpha}_t := \prod_{s=1}^t \alpha_s$.

*Reverse Process* stands as the counter-process, tasked with eliminating the injected noise from the distorted motion sequences. We employ a learnable deep learning model $\bar{\mathbf{x}}_0 = S_\theta(\mathbf{x}_t, t, \mathbf{c})$ to approximate the original sequence. Here, $\mathbf{c}$ signifies the provided condition. In the standard text-driven motion generation, $\mathbf{c}$ is the singular text prompt given. However, for the fine-grained spatio-temporal motion generation task, $\mathbf{c}$ embodies the provided detailed linguistic constraints. The training objective for this network is to minimize the mean square error between the true original sequence $\mathbf{x}_0$ and the estimated one $\bar{\mathbf{x}}0$, which can be formulated as $\mathbb{E}_{x_0,\epsilon,t}[\mathbf{x}_0 - S_\theta(\mathbf{x}_t, t, c)]$.

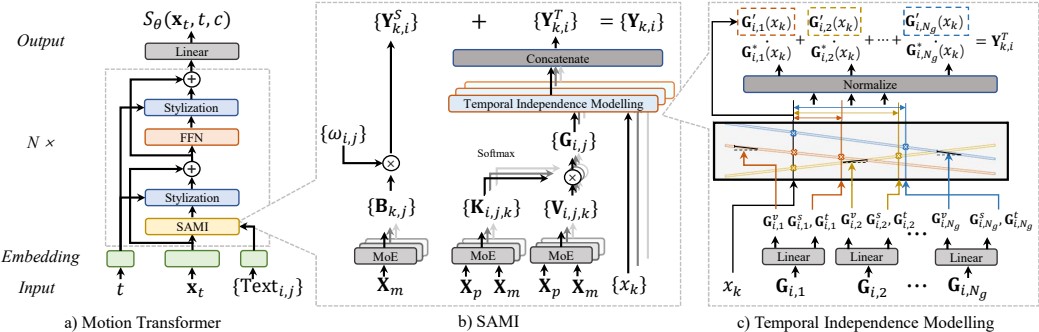

a) Motion Transformer      b) SAMI      c) Temporal Independence Modelling

Figure 3: **Architecutre of FineMoGen**. In Figure a), our employed Motion Transformer first encodes the timestamp $t$, the noise-infused motion sequence $\mathbf{x}_t$, and descriptions $\text{Text}_{i,j}$ into feature vectors, which undergo further processing by $N$ layers. Central to these layers lies our proposed SAMI, which takes into account both spatial independence and temporal independence. In Figure (b)'s left segment, the feature fusion among different body parts is demonstrated. This fusion is driven by a sequence of learnable parameters $\omega_{i,j}$ and the enhanced feature $\mathbf{B}_{k,j}$ derived from MoE projection. The outcome of this spatial modeling is denoted as the feature vector $\mathbf{Y}_{k,i}^S$. Figure b)'s right segment illustrates the process of temporal modeling. Initially, text features and motion features are concatenated and fed into MoE layers, yielding informative features $\mathbf{K}_{i,j,k}$ and $\mathbf{V}_{i,j,k}$. These two feature vectors are then merged into global templates $\mathbf{G}_{i,j}$. For each time moment, the refined feature is obtained from these global templates through temporal independence modeling, as depicted in Figure c). Here, we generate $N_g$ time-varying signals. After normalization by the time differences between $x_k$ and time anchors $\mathbf{G}_{i,j}^t$, we ultimately acquire the refined feature $\mathbf{Y}_{k,i}^T$. Upon summation with $\mathbf{Y}_{k,i}^S$, the final output of SAMI, denoted as $\mathbf{Y}_{k,i}$, is achieved.

*Motion Transformer Network* are frequently chosen as the foundation for estimators in motion diffusion models. These networks consist of several identical blocks that sequentially process the input data as shown in Figure 3. For the provided linguistic prompts, a pre-trained CLIP model [14] is utilized to first extract informative and generalized features. These features are further refined by a series of trainable transformer layers to better align with the requirements of the task at hand. In line with ReMoDiffuse, each block in FineMoGen integrates an efficient mixed attention module and a feed-forward network (FFN). This paper delves into the operational principles of the mixed attention mechanism and subsequently presents a novel attention mechanism specifically designed for spatio-temporal motion composition.

### 3.3 Spatio-Temporal Mixture Attention

In the attention module, we aim to accurately apply fine-grained textual information to the corresponding body parts and time intervals. First we introduce the baseline attention mechanism we used and then illustrate how we modify it to achieve both spatial independence modelling and temporal independence modelling.

**Baseline Mixed Attention.** Here, we follow the design of attention in ReMoDiffuse [21] by incorporating efficient self-attention and efficient cross-attention [16] within the same module, forming a unique Mixed Efficient Attention module (MEA). Suppose the input motion feature sequence and text feature sequence are $\mathbf{X}_m \in \mathbb{R}^{N_m \times L_m}$, $\mathbf{X}_t \in \mathbb{R}^{N_t \times L_t}$ respectively. $N_m$ and $N_t$ are the lengths of motion and text sequences. $L_m$ and $L_t$ are the dimensionality of them. The core idea of MEA is the usage of global templates instead of pair-wise attention. Formally, all elements in $\mathbf{X_m}$ and $\mathbf{X_t}$ first produce a matrix $\mathbf{V} \in \mathbb{R}^{(N_m+N_t) \times H \times L_g}$ by linear projections induced by the weight $W_m^V \in \mathbb{R}^{L_m \times (H \cdot L_g)}$ or $W_t^V \in \mathbb{R}^{L_t \times (H \cdot L_g)}$, where $H$ is the number of heads used in multi-head attention mechanism and $L_g$ is the dimensionality of global templates. Similarly, we get a matrix $\mathbf{K} \in \mathbb{R}^{H \times N_g \times (N_m+N_t)}$ and perform softmax operation on the last dimension, where $N_g$ is the number of global templates in each head. These two matrix are multiplied together to get all global templates $\mathbf{G} \in \mathbb{R}^{H \times N_g \times L_g}$. This process can be written as:

$$\mathbf{V} = [W_m^V \mathbf{X_m}; W_t^V \mathbf{X_t}], \mathbf{K} = [W_m^K \mathbf{X_m}; W_t^K \mathbf{X_t}], \mathbf{G} = \text{Softmax}(\mathbf{K})\mathbf{V}, \qquad (1)$$

where $[\cdot;\cdot]$ indicates a concatenation of two tensors. Matrix $\mathbf{K}$ can be regarded as a series of coefficients that $\mathbf{K}_{i,j,k}$ shows the influence magnitude of $k$th element on $\mathbf{G}_{i,j}$, the $j$th global template in $i$th group. The features in $\mathbf{V}$ are summarized to $H \times N_g$ groups with the normalized weights from $\mathbf{K}$, results in the global templates $\mathbf{G}$.

Similar to standard self-attention, we also produce a matrix $\mathbf{Q} \in \mathbb{R}^{N_m \times H \times N_g}$ to query on the global templates. $\mathbf{Q}_{k,i,j}$ illustrates the significance of $\mathbf{G}_{i,j}$ on the $k$th element. Here we also perform the Softmax operation on the last dimension. and then perform multiplication between $\mathbf{Q}$ and $\mathbf{G}$ to get the refined feature, as the output of this attention module. The process is formulated as:

$$\mathbf{Q} = W_m^Q \mathbf{X_m}, \mathbf{Y} = \text{Softmax}(\mathbf{Q})\mathbf{G}, \tag{2}$$

where $Y \in \mathbb{R}^{N_m \times H \times L_g}$ is the output matrix.

**Temporal Independence.** An intuitive solution is to generate independent refined features for different time intervals and apply them to the corresponding motion segments. However, it lacks a direct mechanism to effectively integrate information across different time intervals and it becomes challenging to generate satisfactory results when we want to generate a varying number of time intervals during testing. The core issue lies in the difficulty of describing the interactions between different time intervals in a scalable manner. To tackle with this issue, we explicitly introduce the concept of time into this procedure. Formally, we propose the following approximation for temporal feature refinement:

$$\mathbf{Y}_{k,i}^T \approx \mu_i(x_k) = \sum_{j=1}^{N_g} \mathbf{G}'_{i,j}(x_k) \cdot \mathbf{G}^*_{i,j}(x_k) \tag{3}$$

where $x_k$ represents the time position of $k$th pose state in the motion sequence. $\mathbf{G}'_{i,j}(x)$ indicates the time-varied signal we derive from the feature vector $\mathbf{G}_{i,j}$ and $\mathbf{G}^*_{i,j}(x_k)$ denotes the relative significance of this template for the $k$th position. Consider the properties of motion generation task, we further construct $\mathbf{G}'_{i,j}(x)$ and $\mathbf{G}^*_{i,j}(x_k)$ as:

$$\mathbf{G}'_{i,j}(x) = \mathbf{G}^s_{i,j} + \mathbf{G}^v_{i,j} \cdot (x - \mathbf{G}^t_{i,j}), \mathbf{G}^*_{i,j}(x_k) = \frac{e^{-(x_k - \mathbf{G}^t_{i,j})^2/\sigma^2}}{\sum_{l \in [1,N_g]} e^{-(x_k - \mathbf{G}^t_{i,l})^2/\sigma^2}}, \tag{4}$$

In this setup, the $j$th global template of the $i$th group is considered as a set of signals propagating outward from the temporal center $\mathbf{G}^t_{i,j}$. We perceive a global template $\mathbf{G}_{i,j}$ as an anchor with its initial state defined as $\mathbf{G}^s_{i,j}$ and velocity represented by $\mathbf{G}^v_{i,j}$. These three matrices are obtained from the original $\mathbf{G}_{i,j}$ through three separate linear projections.

Furthermore, to assimilate influences from all signals, we use the square of time difference as a measure to evaluate the importance of each global template and use a Softmax operation to normalize their weights. A direct advantage of this modeling approach is the ease in appending a new stage following the current one, by adding a bias term to the $\mathbf{G}^t_{i,j}$ accordingly. Therefore, this method proves beneficial in both fully-supervised and zero-shot scenarios.

**Spatial Independence.** In conventional motion generative models [20, 11, 18], the raw motion vectors are directly inputted into a fully-connected layer to generate a latent representation: $\mathbf{F} = W^F(\Theta)$, where $\Theta$ signifies the raw data of a single motion sequence. $\mathbf{F}$ represents the latent representation following the linear projection performed by $W^F$. This operation, while straightforward and effective, often overlooks the distinct interpretations of the motion representation. It also merges data from various body parts, which hampers body part-aware modeling. To overcome this, we manually divide the raw representation into $N_S$ groups, each representing a commonly observed body part. In our work, we consider seven body parts: head, spine, left arm, right arm, left leg, right leg, and trajectory of the pelvis joint. We perform linear projection on each part independently and concatenate them into the latent representation $\mathbf{F}' \in \mathbb{R}^{N_m \times N_S \times L_P}$, where $L_p$ represents the dimensionality of each part. Given our aim for each global template group to correspond exactly to one body part, we set $N_S = N_g$. Additionally, we utilize the Feed Forward Network (FFN) module to process each body part independently.

In addition, we enforce that features from each body part can only contribute to the corresponding group of global templates. In situations where our models are trained without detailed descriptions for each body part, text features from the overall sentence will contribute to each group. However, in the context of the Fine-grained Spatio-temporal Generation task, where we have multiple texts

describing different parts, each text feature sequence will only influence the corresponding body part. This group division approach ensures the preservation of spatial independence.

As a supplement, we introduce the spatial feature refinement here to allow the communication between different body parts in SAMI:

$$\mathbf{Y}_{k,i}^S = \sum_{j=1}^{N_S} \mathbf{B}_{k,j} \cdot \omega_{i,j}, \tag{5}$$

where $\mathbf{B}_{k,j} \in \mathbb{R}^{L_g}$ is the projected feature from $j$th body part of $k$th element. $\omega_{k,j} \in \mathbb{R}^{N_S \times N_S}$ is a learnable parameter to reflect the relative significance. The value of $\omega$ is shared to calculate $\mathbf{Y}_{k,i}^S$ for all $k$ and $i$. It can be explained as a common knowledge to show the correlation between different body parts.

**Sparsely-Activated Mixture-of-Expert** The design of spatio-temporal independence decouples various components of the entire sequence, compelling the model to focus more on local consistency. This introduces certain complexities to the model's learning process. To address this, we attempt to broaden the overall network structure to enhance learning capability. Thus, we employ the Sparsely-activated Mixture-of-Expert technique, which significantly boosts learning effectiveness at the cost of marginal inference time increment. It can adaptively extract informative features from the motion sequence. Specifically, the linear projections we used for calculating $\mathbf{B}_{k,j}, \mathbf{Q}_{i,j}, \mathbf{K}_{i,j}$ from motion features are empowered by MoE. The formulation can be written as:

$$f_{\text{MoE}}(x) = \sum_{i=1}^{N_e} \text{TOP}_k(\text{Softmax}(W_1 x)) \cdot W_2 \phi(W_3 x), \tag{6}$$

where $W_1, W_2, W_3$ are learnable parameters. $\text{TOP}_k(\cdot)$ is a one-hot vector to set all elements to zero except for the largest $k$ ones. Since text features are unchanged during the denoising process, adaptive expression is unnecessary for this part.

**Module Output.** SAMI combines above-mentioned two refined features. The results can be formulated as $\mathbf{Y}_{k,i} = \mathbf{Y}_{k,i}^T + \mathbf{Y}_{k,i}^S$.

### 3.4 Zero-shot Motion Editing

In Section 3.2, we defined a fine-grained description format to precisely control the actions each body part should perform at each time interval. Based on this, we can interactively edit with users through a pretrained Large Language Model (LLM). We use this intermediate format as the current state corresponding to the action, and the user can input natural instructions. The LLM will modify this intermediate format according to the input content to get a new fine-grained description as shown in Figure 2. The motion generation model FineMoGen will then generate a new motion sequence from scratch based on the modified description, serving as the result of the editing. Such a zero-shot pipeline, using the fine-grained description and LLM as a medium, translates the user's natural instructions into content that the model can understand and accept, thereby achieving the effect of editing.

The teaser figure in Figure 1 provides an example of this procedure. It begins with the user providing a detailed description for a two-stage animation. Subsequently, an editing command, such as "I want the person to wave higher and for a longer duration," is issued to modify the motion description for both arms in the second stage. Consequently, the motion in the second stage differs from the previous version. Additionally, the motion in the first stage may also differ from the previous one due to the re-generation process.

### 3.5 Training and Inference

In line with the motion diffusion model literature [21, 18], we randomly mask 10% of the text conditions to approximate $p(\mathbf{x}_0)$. Our training objective is to reduce the mean square error between the predicted initial sequence and the actual ground truth. During the training phase, we generally employ a diffusion process comprising 1000 steps, while a 50-step denoising process is applied at the inference stage.

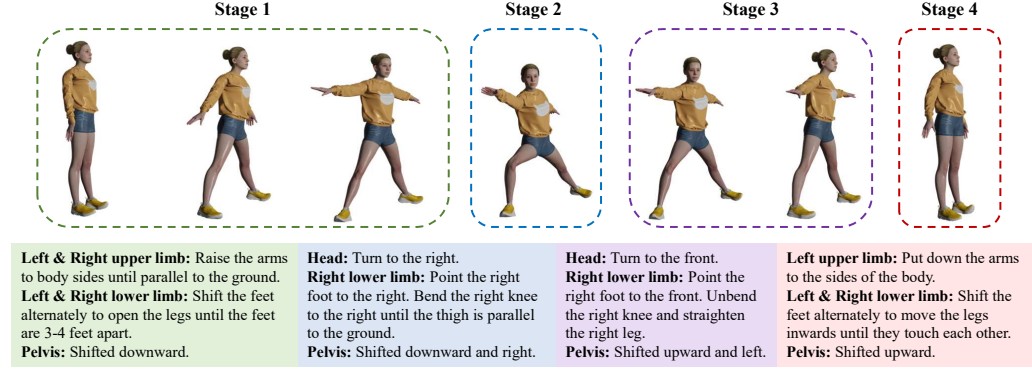

| Stage 1 | Stage 2 | Stage 3 | Stage 4 |
|---|---|---|---|
| **Left & Right upper limb:** Raise the arms to body sides until parallel to the ground. **Left & Right lower limb:** Shift the feet alternately to open the legs until the feet are 3-4 feet apart. **Pelvis:** Shifted downward. | **Head:** Turn to the right. **Right lower limb:** Point the right foot to the right. Bend the right knee to the right until the thigh is parallel to the ground. **Pelvis:** Shifted downward and right. | **Head:** Turn to the front. **Right lower limb:** Point the right foot to the front. Unbend the right knee and straighten the right leg. **Pelvis:** Shifted upward and left. | **Left upper limb:** Put down the arms to the sides of the body. **Left & Right lower limb:** Shift the feet alternately to move the legs inwards until they touch each other. **Pelvis:** Shifted upward. |

Figure 4: **Examples in HuMMan-MoGen dataset**. A motion sequence will be divided into multiple stages. As for each stages, we provide detailed annotation for different body parts.

## 4 Experiments

### 4.1 HuMMan-MoGen Datasets

In addition to commonly used dataset, we re-label the HuMMan [3] dataset, serving as a large-scale benchmark to evaluate Fine-grained Spatio-Temporal Motion Generation quantitatively. The HuMMan [3] dataset includes a total of 500 types of fitness movements, covering the majority of actions we perform in our daily lives across various body parts. We selected 2,968 videos from the 160 types of actions in the HuMMan dataset to be annotated in detail. For each type of action, we first define several standard action phases based on the specifications of that action. We provide detailed descriptions of the movements for each body part during each phase. Specifically, we annotated an overall description, and seven more detailed annotations to describe the head, torso, left arm, right arm, left leg, right leg, and trajectory of the pelvis joint. When annotating each video, we divide the original video into multiple standard action phases. For each phase, we apply the pre-defined fine-grained textual descriptions to accurately describe the spatial and temporal aspects of the captured motion data. Figure 4 show an example of our annotated motion sequence. The total number of annotations reached 102,336. This fine-grained spatio-temporal description provided by our data is not available in other datasets. The broad range of actions covered by this dataset makes it more challenging and valuable as a reference.

### 4.2 Standard Benchmarks and Evaluation Metrics

**Standard Benchmarks.** Apart from our collected HuMMan-MoGen dataset, we also evaluate our proposed pipeline on several leading benchmarks in text-driven motion generation tasks, the KIT-ML dataset[12], the HumanML3D dataset[7] and the Babel dataset[13]. We use these three benchmarks for more comparison.

**Evaluation Metrics.** We employ the same quantitative evaluation measures as used in the literature, including Frechet Inception Distance (FID), R Precision, Diversity, Multimodality, and Multi-Modal Distance. The details of these metrics are discussed in supplementary materials.

### 4.3 Implementation Details

As for all four datasets, we choose almost the same hyperparameters. Regarding the motion encoder, we utilize a 4-layer transformer, with a latent dimension of 7 * 64. Here, 7 corresponds to the number of body parts, and 64 represents the dimensionality for each part. For the text encoder, we build and apply a frozen text encoder found in the CLIP ViT-B/32, supplemented with two additional transformer encoder layers. In terms of the diffusion model, the variances, denoted as $\beta_t$, are predefined to linearly spread from 0.0001 to 0.02, with the total number of noising steps set as $T = 1000$. We use the Adam optimizer to train the model, initially setting the learning rate to 0.0002.

Table 1: **Quantitative results on the HumanML3D test set.** '↑'('↓') indicates that the values are better if the metric is larger (smaller). We run all the evaluations 20 times and report the average metric and 95% confidence interval is. The best result are in bold and the second best result are underlined.

| Methods | R Precision↑ | | | FID↓ | MM Dist↓ | Diversity↑ | MultiModality↑ |
| | Top 1 | Top 2 | Top 3 | | | | |
|---|---|---|---|---|---|---|---|
| Real motions | $0.511^{\pm.003}$ | $0.703^{\pm.003}$ | $0.797^{\pm.002}$ | $0.002^{\pm.000}$ | $2.974^{\pm.008}$ | $9.503^{\pm.065}$ | - |
| Guo *et al.* [6] | $0.457^{\pm.002}$ | $0.639^{\pm.003}$ | $0.740^{\pm.003}$ | $1.067^{\pm.002}$ | $3.340^{\pm.008}$ | $9.188^{\pm.002}$ | $2.090^{\pm.083}$ |
| T2M-GPT [19] | $0.491^{\pm.003}$ | $0.680^{\pm.003}$ | $0.775^{\pm.002}$ | $\underline{0.116}^{\pm.004}$ | $3.118^{\pm.011}$ | $\mathbf{9.761}^{\pm.081}$ | $1.856^{\pm.011}$ |
| MDM [18] | - | - | $0.611^{\pm.007}$ | $0.544^{\pm.044}$ | $5.566^{\pm.027}$ | $9.559^{\pm.086}$ | $\mathbf{2.799}^{\pm.072}$ |
| MotionDiffuse [20] | $0.491^{\pm.001}$ | $0.681^{\pm.001}$ | $0.782^{\pm.001}$ | $0.630^{\pm.001}$ | $3.113^{\pm.001}$ | $9.410^{\pm.049}$ | $1.553^{\pm.042}$ |
| ReMoDiffuse [21] | $\mathbf{0.510}^{\pm.005}$ | $\mathbf{0.698}^{\pm.006}$ | $\mathbf{0.795}^{\pm.004}$ | $\mathbf{0.103}^{\pm.004}$ | $\mathbf{2.974}^{\pm.016}$ | $9.018^{\pm.075}$ | $1.795^{\pm.043}$ |
| Ours | $\underline{0.504}^{\pm.002}$ | $\underline{0.690}^{\pm.002}$ | $\underline{0.784}^{\pm.002}$ | $0.151^{\pm.008}$ | $2.998^{\pm.008}$ | $9.263^{\pm.094}$ | $\underline{2.696}^{\pm.079}$ |

Table 2: **Quantitative results on the KIT-ML test set.**

| Methods | R Precision↑ | | | FID↓ | MM Dist↓ | Diversity↑ | MultiModality↑ |
| | Top 1 | Top 2 | Top 3 | | | | |
|---|---|---|---|---|---|---|---|
| Real motions | $0.424^{\pm.005}$ | $0.649^{\pm.006}$ | $0.779^{\pm.006}$ | $0.031^{\pm.004}$ | $2.788^{\pm.012}$ | $11.08^{\pm.097}$ | - |
| Guo *et al.* [6] | $0.370^{\pm.005}$ | $0.569^{\pm.007}$ | $0.693^{\pm.007}$ | $2.770^{\pm.109}$ | $3.401^{\pm.008}$ | $10.91^{\pm.119}$ | $1.482^{\pm.065}$ |
| T2M-GPT [19] | $0.416^{\pm.006}$ | $0.627^{\pm.006}$ | $0.745^{\pm.006}$ | $0.514^{\pm.029}$ | $3.007^{\pm.023}$ | $\underline{10.921}^{\pm.108}$ | $1.570^{\pm.039}$ |
| MDM [18] | - | - | $0.396^{\pm.004}$ | $0.497^{\pm.021}$ | $9.191^{\pm.022}$ | $10.847^{\pm.109}$ | $\mathbf{1.907}^{\pm.214}$ |
| MotionDiffuse [20] | $0.417^{\pm.004}$ | $0.621^{\pm.004}$ | $0.739^{\pm.004}$ | $1.954^{\pm.062}$ | $2.958^{\pm.005}$ | $\mathbf{11.10}^{\pm.143}$ | $0.730^{\pm.013}$ |
| ReMoDiffuse [21] | $\underline{0.427}^{\pm.014}$ | $\underline{0.641}^{\pm.004}$ | $0.765^{\pm.055}$ | $\mathbf{0.155}^{\pm.006}$ | $\mathbf{2.814}^{\pm.012}$ | $10.80^{\pm.105}$ | $1.239^{\pm.028}$ |
| Ours | $\mathbf{0.432}^{\pm.006}$ | $\mathbf{0.649}^{\pm.005}$ | $\mathbf{0.772}^{\pm.006}$ | $\underline{0.178}^{\pm.007}$ | $2.869^{\pm.014}$ | $10.85^{\pm.115}$ | $\underline{1.877}^{\pm.093}$ |

This learning rate will gradually decay to 0.00002 in accordance with a cosine learning rate scheduler. Training is performed using one Tesla V100, with the batch size on a single GPU set at 128.

As for the HumanML3D and KIT-ML datasets, we follow Guo et al [6], where pose states mainly contain seven different parts: $(r^{va}, r^{vx}, r^{vz}, r^h, \mathbf{j}^p, \mathbf{j}^v, \mathbf{j}^r)$. Here Y-axis is perpendicular to the ground. $r^{va}, r^{vx}, r^{vz} \in \mathbb{R}$ denotes the root joint's angular velocity along Y-axis, linear velocity along X-axis and Z-axis, respectively. $r^h \in \mathbb{R}$ is the height of the root joint. $\mathbf{j}^p, \mathbf{j}^v \in \mathbb{R}^{J \times 3}$ are the position and linear velocity of each joint, where $J$ is the number of joints. $\mathbf{j}^r \in \mathbb{R}^{J \times 6}$ is the 6D rotation of each joint. Specifically, $J$ is 22 in HumanML3D and 21 in KIT-ML. Tuning to the BABEL dataset and our proposed HuMMan-MoGen, we follow TEACH [2] and converts SMPL parameters to a 6D rotation representation together with root translation.

## 4.4 Quantitative Results

We begin by assessing our proposed method on the standard task of text-driven motion generation. As depicted in Table 1 and Table 2, FineMoGen yields results that are competitive with those from existing methods. It's worth noting that while FineMoGen isn't specifically designed for this task, it still achieves satisfactory performance in comparison to other methods.

Table 3 displays the performance of temporal composition on both the BABEL dataset and the HuMMan-MoGen dataset. Our FineMoGen significantly surpasses existing methods in terms of accuracy. This underscores FineMoGen's ability to synthesize motion sequences that are not only more natural but also more aligned with the provided text conditions.

## 4.5 Qualitative Results

Please kindly refer to the supplementary material for qualitative comparison and more demos about both fine-grained motion generation and editing. From the demo video, we can observe the following: 1) Our method demonstrates a better understanding of the provided textual context compared to other approaches. Whether it's single-action generation or continuous action generation, our method generates motion sequences that are more consistent with the text. 2) Due to mathematically coherent temporal modeling, in continuous action generation, the transitions in our generated results are natural, with no apparent artifacts.

Table 3: **Quantitative results of temporal composition on the BABEL test set and HuMMan-MoGen test set.**

| Methods | BABEL | | | | HuMMan-MoGen | | | |
|---|---|---|---|---|---|---|---|---|
| | R Precision↑ | FID↓ | Diversity → | MultiModality↑ | R Precision↑ | FID↓ | Diversity → | MultiModality↑ |
| Real motions | 0.62 | 0.004 | 8.51 | 3.57 | 0.49 | 0.007 | 7.28 | 2.15 |
| TEACH [2] | 0.46 | 1.12 | **8.28** | 7.14 | 0.38 | 1.59 | **7.04** | 6.37 |
| PriorMDM [15] | 0.43 | 1.04 | 8.14 | **7.39** | 0.31 | 1.43 | 6.89 | **6.45** |
| Ours (zero-shot) | 0.51 | 0.84 | 8.01 | 7.28 | 0.41 | 1.34 | 6.87 | 6.41 |
| Ours (fully-supervised) | **0.56** | **0.73** | 7.94 | 7.17 | **0.43** | **1.21** | 6.71 | 6.28 |

Table 4: **Ablation study on HuMMan-MoGen test set.** All methods use zero-shot setting, it means that they are not trained on the temporal composition data.

| Methods | Spatial Dependence | Temporal Dependence | MoE | R Precision↑ | FID↓ | Diversity → | MultiModality↑ |
|---|---|---|---|---|---|---|---|
| Real Motions | - | - | - | 0.49 | 0.007 | 7.28 | 2.15 |
| Baselines | - | - | - | 0.24 | 3.79 | 7.10 | 6.95 |
| | ✓ | - | - | 0.21 | 4.51 | **7.29** | **7.03** |
| | - | ✓ | - | 0.34 | 2.48 | 7.05 | 6.72 |
| | - | - | ✓ | 0.27 | 3.09 | 6.97 | 6.84 |
| FineMoGen | ✓ | ✓ | ✓ | **0.41** | **1.34** | 6.87 | 6.41 |

## 4.6 Ablation Study

Table 4 presents the ablation study conducted on the HuMMan-MoGen dataset. For consistency, we consider only temporal composition in alignment with the results shown in Table 3. The outcomes suggest that both Temporal Dependence and MoE components contribute positively to the overall performance. Conversely, spatial dependence poses challenges for fitting the distribution. We concede that in a temporal composition setting, this module may not be necessary. We also provide ablation analysis on spatial combination in the supplementary material, which indicate consistent conclusions.

## 5 Conclusion and Limitation

In this work, we introduce the task of Fine-grained Spatio-temporal Motion Generation, where more detailed textual descriptions allow users to control and edit the desired actions with greater flexibility. To address this task, we propose a new framework, FineMoGen, which decouples the influence of body parts and time on the feature refinement process in the attention mechanism. This results in superior performance in both fully-supervised and zero-shot scenarios compared to previous works. Given the lack of text-motion datasets specifically designed for fine-grained descriptions, we have compiled a dataset based on the HuMMan dataset. This dataset includes a wide range of actions and an abundance of descriptions segmented by time and body part. These data enable us to analyze and test the effects of different algorithms more effectively.

**Limitation.** While this study proposes a method for spatial decoupling, it is still heavily reliant on a specific fine-grained format, which to a certain extent limits its application scenarios. On the other hand, the efficacy of the motion editing component is strongly tied to the capability of the pre-trained Large Language Model (LLM). In practical use, it sometimes struggles to interpret user requirements accurately. Instruction tuning should be incorporated into the framework in the future, enabling the LLM to better understand instructions related to motion editing.

**Boarder Impact.** The ability to generate fine-grained human motion not only enhances the productivity of the animation industry, but also has potential benefits for helping medical research, improving training athletes and dancers, and providing more realistic interactive experiences in virtual reality. However, it may also be used for illegal activities such as generating fighting motions to create violent videos, or fraud and dissemination of false information.

## Acknowledgment

This study is supported by the Ministry of Education, Singapore, under its MOE AcRF Tier 2 (MOE-T2EP20221-0012), NTU NAP, and under the RIE2020 Industry Alignment Fund – Industry Collaboration Projects (IAF-ICP) Funding Initiative, as well as cash and in-kind contribution from the industry partner(s).

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
