# OpenReview forum: "FineMoGen: Fine-Grained Spatio-Temporal Motion Generation and Editing"
_NeurIPS.cc/2023/Conference — NeurIPS 2023 poster_

### Official Review · Reviewer_4S8a · 2023-06-30

**Soundness:** 2 fair
**Presentation:** 2 fair
**Contribution:** 2 fair
**Rating:** 5
**Confidence:** 4

**Summary:**

Existing methods often struggle to generate intricate motion sequences that align with fine-grained descriptions. Addressing this, the authors introduce FineMoGen, a diffusion-based motion generation and editing framework capable of synthesizing intricate, detailed motions that align with user instructions. FineMoGen employs a diffusion model coupled with a unique transformer architecture called Spatio-Temporal Mixture Attention (SAMI). This architecture optimizes global attention template creation by: 1) explicitly modeling the constraints of spatio-temporal composition, and 2) using a sparsely-activated mixture-of-experts to adaptively extract fine-grained features. In addition, the authors offer the HuMMan-MoGen dataset, designed to facilitate the study of fine-grained motion generation and editing. Extensive experimental validation suggests that FineMoGen outperforms state-of-the-art methods in motion generation quality.

**Strengths:**

1. The authors' contribution of a novel dataset is significant and could stimulate further research in this field. A fine-grained labeled dataset extends the potential for human motion generation applications and aligns with intuitive human approaches to generating and editing human motion.
2. The motivation behind the development of the Spatio-Temporal Mixture Attention model is robust and justified.

**Weaknesses:**

1. The relationship between SAMI and MEA is somewhat unclear, particularly from the perspective of notations. The association between F' and B is not well-explained.
2. The factorization in Equation 4 could benefit from clearer explanation. The current description of G' as a time-varied signal seems insufficient, and the formulation of G* in Equation 4 lacks clarity, with minimal elaboration provided in the main paper.
3. Figure 2 does not adequately represent the primary contributions of this paper. It solely incorporates the LLM module, which, while new, is not the main contribution. The SAMI and MoE, which are crucial to this paper, are omitted.
4. The motivation behind the Sparse-activated Mixture-of-Expert is unclear. I failed to see the inherent correlation between the Sparse-activated Mixture-of-Expert and SAMI.
5. On Line 214, there appears to be a typographical error - "vGiven" should likely be corrected.

**Questions:**

Please refer to weaknesses.

**Limitations:**

The authors mention the limitation and suggest a way to resolve it.

---

> ### Author Rebuttal · Authors · 2023-08-10
>
> We sincerely thank the reviewer for the detailed feedback and comments. We respond to the concerns below:
>
> > **Q1**: The method illustration should be improved, such as the difference between MEA and SAMI, the usage of $G’$, etc.
>
> **A1**: Thanks for your suggestion. We add a new figure in the attached PDF, which clearly shows the detailed structure of our proposed SAMI architecture. We will carefully polish this part in the final version.
>
> > **Q2**: The motivation behind the Sparse-activated Mixture-of-Expert is unclear.
>
> **A2**: The design of spatio-temporal independence decouples various components of the entire sequence, compelling the model to focus more on local consistency. This introduces certain complexities to the model's learning process. To address this, we attempt to broaden the overall network structure to enhance learning capability. Thus, we employ the Sparse-activated Mixture-of-Expert technique, which significantly boosts learning effectiveness at the cost of marginal inference time increment. Our Ablation study corroborates the significance of this aspect. In the final version, we will provide additional elaboration on the motivation behind this approach.
>
> > **Q3**: There are some typos in the current version.
>
> **A3**: We will meticulously review the entire text and rectify any errors in the final version.

---

> > ### Comment · Reviewer_4S8a · 2023-08-18
> > **Post-rebuttal discussions**
> >
> > Dear Authors of Paper 1891,
> >
> > Please accept my apologies for my delayed response. I have carefully read your rebuttal and would like to provide the following comments:
> >
> > 1. **Regarding the difference between MEA and SAMI**: I appreciate the effort you took to illustrate the distinction between these two concepts with a detailed figure. It certainly clarifies the subject in a visually intuitive way. I strongly recommend including this figure in the manuscript to assist the readers.
> > 2. **Regarding the Sparse-activated Mixture-of-Experts (MoE)**: While I understand that previous designs may introduce complexities to the model's learning process, the reasoning behind employing the Sparse-activated MoE technique to "broaden the overall network structure to enhance learning capability" still raises questions for me. Specifically:
> > - ① Could you elucidate the particular features of the Sparse-activated MoE that make it uniquely suited to this problem?
> > - ② Is the "Sparse-activated" design crucial? Is it possible to employ the naive softmax attention instead? Is the concept of mixture-of-experts similar to multi-head in Transformer?
> > - ③ I noticed an inconsistency in terminology between "sparsely-activated" in the abstract and "sparse-activated" in the main text. Please align these terms for clarity, if possible.
> >
> > I look forward to your revisions.
> >
> > Kind regards,
> >
> > Reviewer 4S8a

---

> > > ### Author Response · Authors · 2023-08-21
> > > **Additional Rebuttal**
> > >
> > > We sincerely appreciate your additional suggestions. We intend to delve further into the discussion concerning the Sparsely-activated MoE component.
> > >
> > > > **Q4-1** Could you elucidate the particular features of the Sparse-activated MoE that make it uniquely suited to this problem?
> > >
> > > **A4-1** Our primary objective is to enhance the model's learning capacity as much as possible without significantly increasing testing time. The Sparsely-activated MoE perfectly addresses our dual requirements for speed and accuracy. By leveraging the 'expert' approach, it offers greater flexibility in extracting meaningful information, and during testing, it employs only a minimal number of experts, thus ensuring a modest increase in testing time. To the best of our knowledge, Sparsely-activated MoE could be the optimal choice.
> > >
> > > > **Q4-2** Is the "Sparse-activated" design crucial? Is it possible to employ the naive softmax attention instead? Is the concept of mixture-of-experts similar to multi-head in Transformer?
> > >
> > > **A4-2** We show additional experiments on HuMMan-MoGen test set as below:
> > >
> > > | Method | Number of routing | Number of Experts | R Precision$\uparrow$ | FID$\downarrow$ | Computation Cost |
> > > |:-:|:-:|:-:|:-:|:-:|:-:|
> > > | Real motions | - | - | $0.49$ | $0.007$ | - |
> > > | - | - | - | $0.37$ | $1.95$ | 1x |
> > > | - | 1 | 4 | $0.40$ | $1.57$ | ~1.4x |
> > > | - | 2 | 4 | $0.40$ | $1.49$ | ~2.6x |
> > > | - | 4 | 4 | $0.41$ | $1.41$ | ~5.0x |
> > > | - | 1 | 8 | $0.40$ | $1.54$ | ~1.6x |
> > > | Ours | 2 | 8 | $0.41$ | $1.34$ | ~2.9x |
> > > | - | 4 | 8 | $0.41$ | $1.30$ | ~5.5x |
> > > | - | 8 | 8 | $0.42$ | $1.25$ | ~10.8x |
> > >
> > > The term 'sparsely-activated' does not play a critical role in generating quality. As shown in this table, employing more experts during the testing phase could potentially yield better generation results. Therefore, if precision is the sole consideration, 'sparsely-activated' might not be the optimal choice. However, we also observe that the most noticeable accuracy improvement occurs when transitioning from no MoE usage to using 1 routing, and from using 1 routing to 2 routings. Considering computational complexity, we have opted for a compromise solution.
> > >
> > >
> > > > **Q4-3** I noticed an inconsistency in terminology between "sparsely-activated" in the abstract and "sparse-activated" in the main text. Please align these terms for clarity, if possible.
> > >
> > > **A4-3** Thanks for pointing out this issue. We will revise the entire document and consistently use 'sparsely-activated'.

---

> ### Author Response · Authors · 2023-08-17
> **Follow-up**
>
> Dear Reviewer 4S8a,
>
> Thank you very much again for your time and efforts in reviewing our paper.
>
> As per your suggestion, we have included a new figure to enhance the clarity of the architectural design and have provided a detailed explanation regarding the utilization of MoE.
>
> We just wonder whether there is any further concern and hope to have a chance to respond before the discussion phase ends.
>
> Best regards,
>
> Authors of paper 1891

---

> ### Comment · Area_Chair_oLs7 · 2023-08-17
> **Request for your feedback in light of authors' feedback**
>
> Thank you for your valuable insights and expertise which have contributed significantly to the review process.
>
> Following the initial review, the authors have provided a detailed rebuttal addressing the feedback and comments provided by our esteemed reviewers, including yourself. I kindly request that you take the time to carefully review the authors' rebuttal and assess its impact on your initial evaluation.
>
> Please share your thoughts and any additional points you may have after reading the authors' rebuttal. Thank you very much!

---

### Official Review · Reviewer_CFoF · 2023-07-03

**Soundness:** 3 good
**Presentation:** 3 good
**Contribution:** 3 good
**Rating:** 6
**Confidence:** 3

**Summary:**

The paper proposes a method for diffusion-based motion generation and editing framework for synthesizing fine-grained motions. It takes fine-grained spatio-temporal description. The framework supports zero-shot and fully-supervised generation scenario. The utilize MOE for generating global template. They have also introduced a dataset with fine-grained text annotation. Their method is called spatio-temporal mixture attention (SAMI). To summarize, The fine-grained descriptions are first processed by frozen CLIP followed by two layers of learnable transformer encoders. Next, a text feature matrix is created, and then sent to a diffusion model-based motion generative network to generate corresponding motion sequence.

**Strengths:**

The zero-shot generation is encouraging. The results on HumanML3D and KIT-ML show improved and comparable results with existing methods. The superior temporal composition results show the effectiveness of SAMI. Ablation experiments are a plus.

**Weaknesses:**

The dependencies on the LLM might have two way effect. The natural description of the fine-grained motion may not be best utilized in this scenario.

**Questions:**

Like most other generative methods, it also should have concern for its potential negative impacts. Can some sort of user attribution be incorporated in the model itself?

**Limitations:**

The dependencies on the LLM might have two way effect. The natural description of the fine-grained motion may not be best utilized in this scenario. Like most other generative methods, it also should have concern for its potential negative impacts. Can some sort of user attribution be incorporated in the model itself?

---

> ### Author Rebuttal · Authors · 2023-08-10
>
> We would like to express our sincere thanks for your insightful review and constructive suggestions. We reply to each of your comments here:
>
> > **Q1**: The natural description of the fine-grained motion may not be best utilized in some situations.
>
> **A1**: We concur that employing the generated fine-grained description directly for motion editing might not be the optimal approach in certain scenarios. For instance, a more effective strategy could involve dynamically selecting the appropriate level of detail to ensure enhanced precision during generation. We will try this direction in the future.
>
> > **Q2**: Can some sort of user attribution be incorporated in the model itself?
>
> **A2**: Taking advantage of the diversity within the HuMMan dataset, where each action involves participants of varying ages, genders, and body shapes, our trained models inherently avoid any inductive bias. Furthermore, we will also add a license to regulate users' usage methods in order to mitigate potential negative impacts that the model might bring about.

---

> > ### Comment · Reviewer_CFoF · 2023-08-16
> > **post-rebuttal comments**
> >
> > I thank the authors for responding to reviewers' questions. I am leaning towards accept

---

> > > ### Author Response · Authors · 2023-08-17
> > > **Thanks to Reviewer CFoF**
> > >
> > > Dear Reviewer CFoF,
> > >
> > > We really appreciate you spending the effort to interact with us and provide valuable, constructive feedback about our paper.
> > >
> > > Best regards,
> > >
> > > Authors of paper 1891

---

### Official Review · Reviewer_DsXZ · 2023-07-05

**Soundness:** 3 good
**Presentation:** 3 good
**Contribution:** 3 good
**Rating:** 5
**Confidence:** 3

**Summary:**

The paper targets an interesting problem of fine-grained spatial-temporal motion generation and editing. It introduces a new transformer structure with spatio-temporal mixturee attention. Also, a large scale dataset, called HuMMan-MoGen, has been introduced to train the proposed algorithm. Attractive experimental results have been reported in the paper.

**Strengths:**

1. A new large-scale dataset with fine-grained spatio-temporal descriptions have been introduced in the paper.

2. Attractive performance has been reported on the motion generation benchmark.

3. The problem of fine-grained motion generation and edting would be helpful for the industry.

**Weaknesses:**

1. The proposed HuMMan-MoGen dataset is originally from the HuMMan dataset. Is it possible to the dataset convering sufficient diversity of the finegrained motion clips?

2. The annotations of different groups would make it easy to represent the motions. But some motions have been defined as a whole body rather than a combination of motions from different body parts. Is there any potential negative impact on this problem?

3.  I would suggest to include the dataset contruction process in the paper rather than leaving it in the supplementary.

**Questions:**

Please address the questions in the weakness section. Also, will the dataset be public released together with the finegrained description?

**Limitations:**

The paper already discussed the limitations in the paper.

---

> ### Author Rebuttal · Authors · 2023-08-10
>
> We sincerely appreciate your constructive and helpful suggestions. And we carefully reply to each of the reviewer's comments below.
>
> > **Q1**: Is HuMMan-MoGen diverse enough?
>
> **A1**: The HuMMan dataset comprises 500 actions, strategically curated to encompass foundational movements. From this collection, our proposed HuMMan-MoGen carefully selects the most representative 200 categories. These categories encompass a wide range of movement types for each body part. Notably, our dataset is intentionally structured to facilitate spatio-temporal composition, enabling models trained on it to generate an extensive variety of motion clips.
>
>
> > **Q2**:  Some motions have been defined as a whole body rather than a combination of motions from different body parts. Is there any potential negative impact on this problem?
>
> **A2**: Currently, as for each motion sequence, we provide an overall text description and detailed descriptions for seven different body parts. Taking running as an example, we will provide a holistic description such as 'a person is running', as well as finer-grained descriptions for each body part. These detailed descriptions illustrate how the limbs move and coordinate with each other during the running process. Therefore, for such comprehensive descriptions, fine-grained descriptions can serve as supplementary information, offering richer details for the model to reference.
>
> > **Q3**: The dataset construction process should be included in the paper.
>
> **A3**: Thanks for your suggestion. We will refine this part in our final version.
>
> > **Q4**: Will the dataset be public released together with the finegrained description?
>
> **A4**: Yes, we will publish all the SMPL sequences along with their corresponding fine-grained descriptions upon acceptance.

---

> > ### Comment · Reviewer_DsXZ · 2023-08-14
> >
> > The rebuttal addressed most of my concerns and I will keep the postive view on the paper.

---

> > > ### Author Response · Authors · 2023-08-17
> > > **Thanks to Reviewer DsXZ**
> > >
> > > Dear Reviewer DsXZ,
> > >
> > > We really appreciate your prompt response and highly valuable feedback.
> > >
> > > Best regards,
> > >
> > > Authors of paper 1891

---

### Official Review · Reviewer_C9Az · 2023-07-06

**Soundness:** 3 good
**Presentation:** 2 fair
**Contribution:** 3 good
**Rating:** 6
**Confidence:** 4

**Summary:**

The paper introduces a compositional text-to-motion synthesis approach that allows for control up to body part level. For fine-grained spatio-temporal editing, the motion of each body part from a predetermined set (e.g., head, arms, legs, etc.) can be described. Furthermore, for zero-shot motion control, the model has an interface with LLMs to support high-level motion descriptions, which are converted to part-specific commands. The paper introduces FineMoGen, a diffusion-based model with novel spatio-temporal mixture attention (SAMI). Specifically, each body part is represented with a spatial attention block and conditioned only on the corresponding part’s text feature to ensure spatial independence. Similarly, the temporal block aims to make smooth transitions between different stages (i.e., different atomic motions). The proposed model performs competitively on the ***atomic text-to-motion benchmarks such as HumanML3D and KIT-ML. In the ***compositional text-to-motion task, it performs quantitatively better than the baselines on the BABEL and the proposed HuMMan-MoGen datasets. There are major artifacts in the synthesized motions (e.g., implausible movements and foot skating), though.

*** I distinguished two tasks by tagging them as "atomic" and "compositional". While the former focuses on generating a short motion snippet from a single instruction describing an atomic event, the latter also involves transitions between semantically different motion descriptions, requiring the composition of several instructions.


**Strengths:**

**originality**
The problem setting is interesting and practical. The proposed model is arguably novel. Ideas from the prior works are nicely extended.

**quality**
The paper is well-motivated, and the concerns are addressed in the design of the proposed model. The decoupling of the spatial and temporal blocks and the corresponding inputs (i.e., part-specific text commands and descriptions for different stages, respectively) seems to be effective.

**clarity**
Although the presentation of the technical details could be improved, it is straightforward to get an overview of the motivation and the proposed approach.

**significance**
The proposed model performs well in evaluations in both the atomic text-to-motion and compositional text-to-motion tasks by allowing for fine-grained editing. I believe this is an interesting direction and could be valuable for the community.


**Weaknesses:**

* The motion quality of the synthetic samples is one of the major weaknesses of this work. While the motion samples in the first part of the supplementary video (examples 1, 2, and 3) are okay, the rest, including the qualitative comparison and the editing sections, have major artifacts. Unfortunately, the inference settings (i.e., the dataset, type of text commands, etc.) are not provided. (**Q1**) Could the authors explain why this is the case?


* (**Q2**) Similarly, the paper does not provide details on the underlying motion representation. Is it 3D positions or rotations? If the latter, which representation? What do the "raw motion vectors" look like?

* I find it not straightforward to follow the technical details in Section 3.3. At least the transition from the "Efficient Attention" paragraph to Section 3.3 was not smooth. I think a revision would help. Moreover, an intuitive figure could be highly useful for conveying the message.


**Questions:**

I have listed my questions regarding the weaknesses of the paper above (Q1 and Q2). Here I am raising additional questions for further clarification.

**Q3**) It is not clear to me how interactive editing works. Is the motion initialized with the output of the previous pass or generated from scratch as an additional stage?

**Q4**) The supplementary video does not have any results for the atomic text-to-motion task. I am curious if the artifacts also exist in this task. Could the authors provide a qualitative comparison with the baselines?


**Limitations:**

Yes, the paper briefly mentions limitations and the broader impact.

---

> ### Author Rebuttal · Authors · 2023-08-10
>
> We sincerely appreciate your recognition of our work’s strengths and insightful suggestions. Here we respond to each of your comments in detail below.
>
> > **Q1**: The inference setting is not clear. The qualitative results have artifacts.
>
> **A1**: The initial examples represent the ground truth data from our HuMMan-MoGen dataset, while the subsequent ones are all generated by algorithms applied to the BABEL dataset. It is acknowledged that generating fine-grained motion remains a challenging endeavor. Despite our method exhibiting a significant lead over existing approaches, the generated results are not flawless. We are committed to persistently pushing the boundaries of this task.
>
> > **Q2**: The details of motion representation should be clarified.
>
> **A2**: Thanks for your suggestions. As for the HumanML3D and KIT-ML datasets, we follow Guo et al[1], where pose states mainly contain seven different parts: $(r^{va},r^{vx}, r^{vz},r^h, \mathbf{j}^p, \mathbf{j}^v, \mathbf{j}^r)$. Here Y-axis is perpendicular to the ground. $r^{va},r^{vx}, r^{vz} \in \mathbb{R}$ denotes the root joint's angular velocity along Y-axis, linear velocity along X-axis and Z-axis, respectively. $r^h \in \mathbb{R}$ is the height of the root joint. $\mathbf{j}^p, \mathbf{j}^v \in \mathbb{R}^{J \times 3}$ are the position and linear velocity of each joint, where $J$ is the number of joints. $\mathbf{j}^r \in \mathbb{R}^{J \times 6}$ is the 6D rotation of each joint. Specifically, $J$ is 22 in HumanML3D and 21 in KIT-ML. Tuning to the BABEL dataset and our proposed HuMMan-MoGen, we follow TEACH[2] and converts SMPL parameters to a 6D rotation representation together with root translation. We will add these details in the final version.
>
> \[1\] Guo et al. Generating Diverse and Natural 3D Human Motions from Text
>
> \[2\] Athanasiou et al. TEACH: Temporal Action Composition for 3D Humans
>
> > **Q3**: How does interactive editing work is not clear.
>
> **A3**: Since the modified segments could potentially impact the continuity, after each description modification, we refrain from using the previous generated results. Instead, we regenerate a new sequence from scratch. We will add more implementation details about this part in the final version.
>
> > **Q4**: Qualitative results in the atomic text-to-motion generation task.
>
> **A4**: We provide the link for a new demo video to AC. We qualitatively compare our method with MDM, MotionDiffuse and ReMoDiffuse. Our results are similar to ReMoDiffuse and outperform the other two. We hope this video can address your concern.
>
> > **Q5**: Method illustration should be improved.
>
> **A5**: Thanks for your suggestion. We add a new figure in the attached PDF. We will carefully polish this part in the final version.

---

> > ### Comment · Reviewer_C9Az · 2023-08-16
> > **Post-rebuttal Comment**
> >
> > I thank the authors for their rebuttal. My concerns are addressed. I read other reviews as well. It looks like no other major concerns have been raised. I am lenient towards acceptance. I'll revise my score after the reviewer discussion period.

---

> > > ### Author Response · Authors · 2023-08-17
> > > **Thanks to Reviewer C9Az**
> > >
> > > Dear Reviewer C9Az,
> > >
> > > We sincerely appreciate the comments and raising the score of our paper.
> > >
> > > Best regards,
> > >
> > > Authors of paper 1891

---

### Author Rebuttal · Authors · 2023-08-10

We express our gratitude to all the reviewers for their valuable and constructive feedback.  We are glad at reviewers’ recognition of our work's strengths:

**1)** The problem setting is interesting and practical (R-C9Az, R-DsXZ, R-CFoF, R-4S8a).

**2)** The decoupling of spatial independence and temporal independence is interesting and achieves impressive results. (R-C9Az, R-DsXZ, R-CFoF, R-4S8a).

**3)** The collected dataset is significant to the community (R-DsXZ, R-4S8a).

In the accompanying file, we have included a new figure to enhance the clarity of the architectural intricacies in our proposed method.

---

### Decision · Program_Chairs · 2023-09-21

**Decision:**

Accept (poster)

**Comment:**

All reviewers find that the paper is well-motivated and the proposed method is shown to be effective. They appreciate the new large-scale dataset with fine-grained spatio-temporal descriptions that was introduced in the paper.

In the initial round of review, there are concerns about the motion quality of the synthetic samples, not sufficient details on the underlying motion representation, the dependencies on the LLM, unclear relationship between SAMI and MEA, and whether the "Sparse-activated" design is crucial. The authors addressed all these major concerns in the rebuttal period, which is acknowledged by all the reviewers. Therefore, we would like to suggest acceptance of the paper.

It is requested that the authors address all the requests in the ethics reviews. They are also suggested to include all the promised revisions during rebuttal period in the final version.